# Students' Perceptions of Active Learning Classrooms from an Informal Learning Perspective: Building a Full-Time Sustainable Learning Environment in Higher Education

**Lei Peng** [1,2] , **Shitao Jin** [1,2,*] , **Yuan Deng** [1,2] and **Yichen Gong** [3]

1   School of Architecture & Urban Planning, Huazhong University of Science and Technology, Wuhan 430074, China; penglei@hust.edu.cn (L.P.); m202073637@hust.edu.cn (Y.D.)
2   Hubei Engineering and Technology Research Center of Urbanization, Wuhan 430074, China
3   Department of Mathematics, New York University, Brooklyn, NY 10012, USA; yg2082@nyu.edu
*   Correspondence: shitaojin@hust.edu.cn; Tel.: +86-189-8601-1150

**Abstract:** Under the influence of education for sustainable development, active learning classrooms, as the new learning environment in higher education, have become increasingly diversified and flexible, with a greater emphasis on student experience and engagement. Most research on these learning environments is centered on formal learning analysis and discussion, whereas empirical research on the perception of informal learning in active learning classrooms outside of class time is lacking. Based on informal learning perceptions, this study explored the strengths and weaknesses of active learning classroom spatial environments using a mixed quantitative and qualitative research approach. Through a questionnaire survey of 154 students and one-on-one semi-structured interviews with 15 students, this study found that most students were satisfied with the informal learning experience in active learning classrooms, comfortable and flexible space perception and humanized learning support facilities in active learning classrooms were critical spatial factors influencing students' informal learning, the private environment and positive learning atmosphere in active learning classrooms could promote students' informal learning behaviors, and for active learning classrooms, better resource management could help them develop a better quality full-time learning environment. Based on these findings, this study makes recommendations for optimizing the environment design and management of active learning classrooms.

**Keywords:** sustainable education; higher education; learning environments; active learning classrooms; informal learning; student perceptions

## 1. Introduction

The concept and practice of education for sustainable development and learning environments are inextricably linked. As the physical entity of sustainable education implementation, learning environments carry the development of educational methods, learning concepts, resources and facilities, and cultural cognition [1]. As a dominant learning environment, the development of the classroom is inevitably influenced by numerous driving variables in sustainable education, such as personalized and adaptable learning capacity, learner perception and engagement, and the active learning pedagogical practices involved [2]. Along with the development of sustainable education, the innovation of learning concepts, and the emergence of new technologies, an increasing number of researchers have begun to promote the innovation and practice of classroom space, particularly in the United States, the United Kingdom, Australia, and some education developed countries [3–6], various more innovative and flexible types with an emphasis on active learning attaching great importance to the students' experience and participation in active learning classrooms have emerged.

Active learning classrooms are student-centered learning environments supported by digital technology and interactive information integration [7] which not only meet the dual needs of constructivism teaching implementation and innovative practice application [8] but also meet the needs of personalized and adaptive multi-oriented training [9]. There are three driving factors behind this. First, both the goals of sustainable education and the development of learning science theory emphasize the importance of active learning skills (e.g., concentration, teamwork, self-regulation, and social adaptation) in the cultivation of future higher education talent [10–12], while the spatial environments that support student-centered and interdisciplinary collaboration are more conducive to the development of students' active learning skills [13]. Second, the deep integration of information technology and educational concepts enables generating and disseminating knowledge in novel ways. Active learning classrooms that mix technological devices and classroom activities blur the barriers between physical and virtual spaces, promoting a learning environment that is flexible in time and sustainable in space [14]. Third, the promotion of lifelong learning [15] and the establishment of the informal learning paradigm [16] have provided learners with a more diversified and complex personalized learning environment, requiring a richer continuum of learning spaces to meet learners' requirements.

As a new field of development, active learning classrooms are attracting many researchers and practitioners, and existing research focuses on the following two aspects: (1) the description and construction of educational concepts and technical facilities and the spatial design of active learning classrooms and (2) teaching effectiveness and classroom evaluation of active learning classrooms. However, active learning classrooms, as an important carrier of sustainable education development and an important part of the spatial continuum constructed by learning theory, still have gaps in the perception research and effect evaluation of extracurricular students' active knowledge construction, informal learning participation, and personalized learning activities. At present, the construction of active learning classrooms in various universities has increased significantly. For example, Auburn University has built more than 50 new active learning classrooms in 6 years [17], Sichuan University has renovated and built more than 400 active learning classrooms since 2012 [18], and Huazhong University of Science and Technology has also built and put into use 101 active learning classrooms from 2018 to now [19], and a large number of active learning classrooms have become one of the main spatial places for students' informal learning activities. Research on the perceptual experience of unstructured learning activities for such structured environments will contribute to the sustainable development of active learning classrooms by providing better quality opportunities for knowledge construction, collaborative exchange, and practical application to maximize support for high-quality and full-time learning. This study will concentrate on students' spatial perceptions of active learning classrooms from the perspective of informal learning and will utilize empirical research methods to investigate the impact of active learning classrooms on college students' experiences and perceptions of informal learning activities in order to improve the sustainability of active learning classrooms and provide a reference basis for optimizing the design and management of learning spaces. Based on this, this study proposes the following three research questions:

1. Are students satisfied with their informal learning experiences in active learning classroom environments?
2. What are the critical spatial factors in the active learning classroom environments that can influence students' informal learning experiences?
3. How can active learning classrooms be further improved and optimized to make them more sustainable learning environments?

## 2. Literature Review and Research Framework

### 2.1. Theoretical Research and Practical Exploration of Active Learning Classrooms

Active learning classrooms are not isolated physical environments but rather a carrier and expression of the constructivist learning concept and learning space continuum, imply-

ing a shift from the traditional teacher-led paradigm of knowledge transfer to a student-led paradigm of construction and problem solving [6,20]. Furthermore, active learning classrooms should not be viewed simply as a new teaching environment, as teaching is simply one of its purposes. The learning environment also implies a shift toward sustainable learning activities, where learning does not only take place in a specific classroom but is a continuous and open process [21]. Active learning classroom environments allow for richer learning expressions, such as self-study, group activities, and workshops.

Theoretical research on active learning classrooms includes exploring design principles and the analysis of empirical evaluations. In terms of design principles, the pedagogy-space-technology framework developed by Radcliffe et al. [22] explains the leading theoretical principles of active learning classrooms. Pearshouse et al. [23] illustrated the relationship between learning environments and education through the why-what-what framework. In terms of empirical evaluation, it primarily investigates the practical impact of different new teaching techniques [24,25], advanced communication and information technologies [26,27], and redesigned innovative space forms used in active learning classrooms [6,28]. Simultaneously, organizations and national teams such as the UK Higher Education Funding Council [29], Australia's Next Generation Learning Environment Project [30], and the United States' National Learning Infrastructure Initiative (NLII) have systematically sorted out and established corresponding learning space evaluation guidelines.

The practical exploration of active learning classrooms began in the early 21st century with the Student-Centered Active Learning Environment for Undergraduate Program (SCALE-UP) [31] proposed by North Carolina State University, which formed the foundation for active learning classrooms with a student-centered teaching and learning environment [32]. MIT's Technology Enables Active Learning (TEAL) space [7] improves student cooperation and learning by utilizing more complex visual media simulations and personal response systems. The University of Minnesota introduced the Pedagogy-rich, Assess learning impact, Integrate innovations, Revisit emerging technologies (PAIR-up) active learning classroom [33] in 2006, increasing the flexibility of the space and allowing teachers and students to experience a new classroom design and variety of teaching strategies. The University of Iowa's Transform, Interact, Learn, Engage (TILE) classroom spaces [34] combine faculty instructional strategies with the design of classroom spaces and offer a greater diversity of technology equipment and classroom sizes. In 2015, Thomas Jefferson University built several active learning classrooms of various shapes and sizes, collectively known as "Nexus Learning Hubs", which offer a variety of furniture configurations and maximize each student's workspace, creating many collaborative group environments [35]. Auburn University designed and built more than 50 active learning classrooms, named Engaged, Active Student Learning (EASL), from 2011 to 2017. The university has specially set up management and maintenance departments for these classrooms and cultivated a number of diversified teaching staff suitable for these classrooms to encourage students' active participation [17]. Huazhong University of Science and Technology (HUST) commissioned 101 active learning classrooms in 2018, known as "smart classrooms", which incorporate technologies such as the Internet of Things (IoT), multi-screen interaction, and smart interactions, as well as more comfortable furniture and interior decoration to meet innovative teaching and learning developments [19].

In general, prior research has made many important attempts from many perspectives toward active learning classrooms, but there is still potential for exploration in two areas. First, most studies begin with the perspectives and expectations of designers and educators, with relatively little research on students' experiences and perceptions of actually using active learning classrooms, while research on students' perceptions of this new learning space based on their motivation and learning experiences is a critical element for the further development of active learning classrooms. Second, existing studies mainly focus on the research and evaluation of formal learning in active learning classrooms, whereas there is a relative lack of research on students' extracurricular learning activities in active learning classrooms. As for active learning classrooms, most of the time, they are open for

informal learning activities. How to make it a truly full-time sustainable education learning environment needs a more comprehensive perspective.

### 2.2. Informal Learning and Its Environmental Impact

Informal learning is an active learning behavior that is self-initiated, self-regulated, and self-responsible by learners outside of formal school education. Its knowledge comes from learning diversity, which is a kind of integrated learning with a social nature [36]. American adult educator Victora J. Marsick defines informal learning as a de-structured form of learning with a sense of active learning closely related to the surrounding environment and social conditions [16]. At present, research on informal learning mainly focuses on learners' experiences and perceptions, and its findings can visually reflect learners' preferences and effects on informal learning. Among this research, qualitative research methods include one-on-one structured interviews [37], focus group interviews [38], and behavioral logs [39], while quantitative research methods include questionnaires [40], behavioral measures [41], and environmental preference surveys [42], and mixed research methods include a combination of questionnaires and interviews [43,44], as well as a combination of delayed photography and focus group interviews [45]. In general, there are many examples in the literature in the study of informal learning that use qualitative or quantitative research methods, but there are few examples in the literature that use mixed research methods.

Unlike formal learning, informal learning emphasizes the initiative, diversity, and randomness of students' learning, so the design of its learning environment should also focus more on students' self-direction and multi-dimensional interactive experiences [46] in order to promote communication and cooperation, achievement demonstration, and learning participation in informal learning activities. Research on the design of informal learning environments can currently be summarized by the following four points: first is the comfort of the indoor environment, such as indoor lighting [38], ventilation, noise [37,47], temperature [48], materials, and color [49], second is a flexible spatial layout, such as flexibility of furniture [49], diversity of spatial layout, and the openness and privacy of the space [50], third is a positive spatial atmosphere, such as a good learning atmosphere [36], better spatial accessibility, rich spatial level, and humanized infrastructure [51], and fourth is modern electronic facilities, such as interactive whiteboards [52], wireless microphones, and interactive visual software. Because of the social character of informal learning in higher education, research at this stage has focused on public spaces on campus, such as libraries, student cafeterias, dormitory buildings, and other potential outdoor public places. However, with the emergence of active learning classrooms, their high-quality learning environment has attracted many students to use them for informal learning outside of the curriculum. Therefore, there is an urgent need to investigate the impact of the environmental design of this learning space on students' experiences and perceptions of informal learning.

To summarize, active learning classrooms, as a new generation of learning environment, can not only meet formal learning needs but also promote students' free and flexible informal learning, encourage students to think and solve problems independently, and cultivate students' sense of cooperation and social awareness. However, how to optimize and design "full-time learning catalysts" such as active learning classrooms with an informal learning perspective has become an important research issue in current and future sustainable learning environments in higher education.

### 2.3. Research Framework

Based on the above research, we can learn that the learning environment in higher education is one of the main factors affecting students' informal learning experiences and perceptions and that a better-quality learning environment can promote more active informal learning. This study focused on the learning environment of active learning classrooms to investigate the influential relationship between students' informal learning experiences and the perception of space in active learning classrooms. As shown in

Figure 1, a blended research approach was used to obtain students' satisfaction with the informal learning experience in the existing learning environment and the critical spatial elements that can enhance and improve students' informal learning in order to propose corresponding recommendations and strategies for the subsequent optimization and design of active learning classrooms as sustainable educational environments.

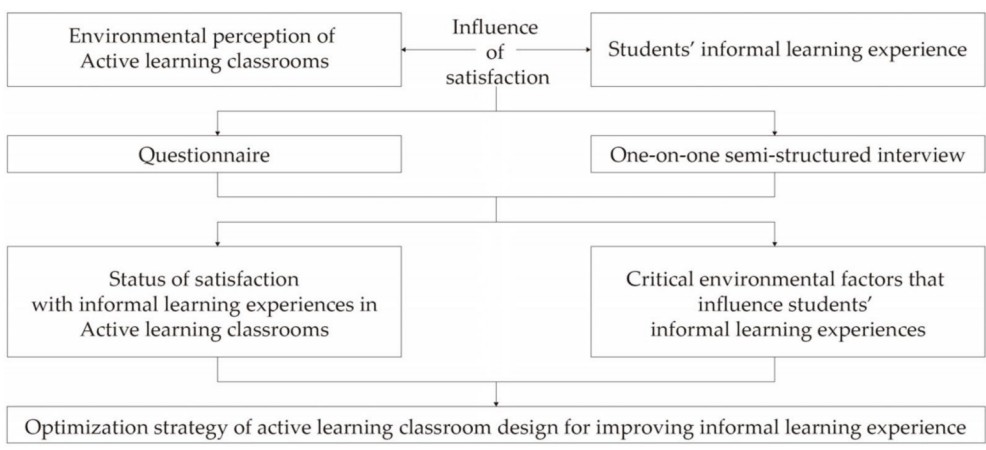

**Figure 1.** Research framework.

## 3. Research Methodology

### 3.1. Research Design

This study used a student-oriented questionnaire and semi-structured interviews to obtain empirical data on the effects of active learning classroom environments on college students' perceptions of informal learning. The combination of quantitative and qualitative research methods has the following advantages: first, it can reveal differences between individual and group responses, which helps to improve the study's explanatory and persuasive power [53], and second, it can construct a more comprehensive understanding and perception of the learning environment by the respondents, which helps to improve the findings' generalizability and practicability.

The research subjects selected for this study were active learning classrooms at HUST. These research subjects were selected for the following reasons: the active learning classrooms at HUST are arranged following the student-centered design principle advocated by SCALE-UP and TEAL, and the environment design and technology updates have been carried out to a certain extent according to the development of higher education and information technology. This can reflect a learning environment concerned with sustainable education development. In addition, HUST built and used more than 110 active learning classrooms in 2018, all of which are designed with different types of spatial features based on the demands of professional courses and disciplines, and the rich sample size and diverse spatial environments can be sufficiently representative. The active learning classrooms in this school can be divided into four types according to their building size, space layout, and furniture design (Table 1): the table and chair integrated type, individual splicing combination type, multi-person splicing combination type, and multi-person fixed combination type. Finally, this university's active learning classrooms are used on a daily basis to take a large number of general and specialized subject courses, and a large number of students study informally in the classrooms outside of class time, so their layout and usage patterns are abundant and varied to meet the individual needs of different learning activities. As shown in Table 2, in HUST, the table and chair integrated type of classroom is small in size, and their centripetal and parallel layouts are mostly used for small-class teaching, while the free layout is mostly used for students' extracurricular group activities. The centripetal and parallel layouts of the individual splicing combination type of classrooms can be used for general and specialized courses, while the free layout is mostly used for individual and independent learning of students. All three usage layout modes of

the multi-person splicing combination type of classroom can meet the needs of the course, and their free and centripetal layout can be used for informal learning discussions and extracurricular communication among students. The multi-person fixed combination type of classroom is more versatile in its use due to the fixed layout, and the electronic monitors that combine with each multi-person desk make them the primary learning venue for student debriefing and discussion outside of class.

**Table 1.** Four types of active learning classrooms on the HUST campus.

| Classroom Type | Table and Chair Integrated | Individual Splicing Combination | Multi-Person Splicing Combination | Multi-Person Fixed Combination |
|---|---|---|---|---|
| Classroom size | 30 people | 42 people | 48 people | 40 people |
| Space layout |  |  |  |  |
| Furniture design | Movable integrated furniture (with storage areas) | Movable fan-shaped desks Movable seats | Multi-person movable trapezoidal desks Movable seats | Multi-person fixed U-shaped desks (with power outlets and storage areas) Movable seats |
| Per capita area | 1.9 m$^2$/person | 2 m$^2$/person | 1.8 m$^2$/person | 2.2 m$^2$/person |
| Technical equipment | Display (*4) Movable whiteboard Desktop computer Control terminal Microphone | Display (*6) Movable whiteboard Desktop computer Control terminal Microphone | Display (*6) Movable whiteboard Desktop computer Control terminal Microphone | Display (*6) Movable whiteboard Desktop computer Control terminal Microphone |
| Photos |  |  |  |  |

The investigation of this study was conducted in the autumn semester of 2021, and the questionnaires and semi-structured interviews were conducted at the same time and continued for 3 months. The specific research content of this study can be divided into two phases. The first phase was a quantitative study in which the active learning classrooms at HUST were visited during non-class periods and questionnaires were distributed to the students in the classrooms who were engaged in informal learning activities such as extracurricular independent learning or extracurricular group learning, after which the collection and related analyses were conducted. The second phase was a qualitative study, which involved validation and extension of the results of the quantitative analysis. To ensure uniformity of the findings, the interviewees were recruited from the students who participated in the questionnaire and engaged in one-on-one semi-structured interviews in active learning classrooms. During the interview, the interviewees' thoughts were not influenced. They could evaluate the learning environment according to their own

experiences and perceptions to obtain more accurate and reliable research data. The consent of the participating students was obtained before the questionnaire survey and the semi-structured interviews began.

**Table 2.** Usage of 4 active learning classroom types at HUST.

| | Usage 1 | Usage 2 | Usage 3 |
|---|---|---|---|
| Table and chair integrated | Parallel layout<br>Traditional teacher teaching<br>(small classes) | Centripetal layout<br>Student group discussion | Free layout<br>Extracurricular self-study,<br>team activities |
| Individual splicing combination | Parallel layout<br>Traditional teacher teaching | Centripetal layout<br>Student group discussion | Free layout<br>Extracurricular<br>personal learning |
| Multi-person splicing combination | Parallel layout<br>Traditional teacher teaching | Centripetal layout<br>Student group discussion | Free layout<br>Extracurricular team activities |
| Multi-person fixed combination | Centripetal layout<br>Traditional teacher teaching | Centripetal layout<br>Student group discussion | Centripetal layout<br>Student extracurricular<br>team report |

### 3.2. Questionnaire and Interview Design

The questionnaire used in this study was divided into two parts. The first part asked students to provide their current active learning classroom numbers and personal information, including gender, academic stage, and subject major. The second part was a Likert scale to assess the impact of the active learning classroom environments on students' perceptions of informal learning activities. The dimensions of this scale were based on Lei et al. [54] 's questionnaire "Exploring Active Learning Classroom Space Factors Affecting Students' Learning Experiences", which classified classroom environments into four dimensions: "Instructional Interaction", "Furniture Perception", "Learning Support", and

"Physical Environment", and it has been shown to have a significant impact on students' learning experiences in each of these four dimensions. The questions of the scale were based on the "Classroom Environment Quality (IEQ) Survey Scale" by Choi et al. [55] and the "Influence of Student Performance in Active Learning Classroom Environment Scale" by Yang et al. [56]. While extracting the mature scale, this study also combined the spatial attributes and characteristics of the current active learning classrooms at HUST, as well as the behavioral characteristics of students' informal learning activities to compile each specific dimension and item of the questionnaire. The final redesigned scale contained 22 items, including 8 questions on instructional interaction, 4 questions on furniture perception, 3 questions on learning support, and 7 questions on the physical environment, all analyzed using a 5-point Likert scale (Appendix A). Statistical software SPSS 24 was used as the main tool to analyze the questionnaire, the specific data analysis process can be found in the Supplementary Materials. A total of 160 questionnaires were distributed in this study. After excluding incomplete, repeated, or invalid questionnaires, 154 valid questionnaires were finally used for analysis, and the questionnaire return rate was 96.2%. Table 3 shows the sample distribution of students in this survey. Because HUST is an institution of higher education with a focus on science and engineering, most students in this university study science and engineering disciplines, and fewer students study natural and social sciences, so the sample distribution of students in this study is basically consistent with the overall distribution of students' attributes such as gender, major, and grade level at this university. In addition, the sample size of the students participating in this survey was statistically significant compared with the number of specific items on the scale [57]. The internal consistency test was conducted on the questionnaire data, and Cronbach's alpha value was 0.939, indicating that the reliability of this questionnaire was good.

**Table 3.** Sample distribution of students participating in the questionnaire (N = 154).

| Classroom Type | Number | Percentage |
|---|---|---|
| Table and chair integrated | 31 | (20.12%) |
| Individual splicing combination | 103 | (66.88%) |
| Multi-person splicing combination | 10 | (6.49%) |
| Multi-person fixed combination | 10 | (6.49%) |
| **Gender** | **Number** | **Percentage** |
| Male | 113 | (73.37%) |
| Female | 41 | (26.62%) |
| **Academic Stage** | **Number** | **Percentage** |
| Undergraduate | 125 | (81.16%) |
| Masters | 19 | (12.33%) |
| PhD | 10 | (6.49%) |
| Other | 0 | (0.00%) |
| **Professional Disciplines** | **Number** | **Percentage** |
| Philosophy, economics, and law | 5 | (3.24%) |
| Education, literature, and history | 6 | (3.89%) |
| Science, engineering, agriculture, and medicine | 142 | (92.20%) |
| Military science, management science, and art | 1 | (0.64%) |

The interviews used in this study included the following topics (Appendix B). First, how often do students come to active learning classrooms for informal learning? Second, in what ways do students engage in informal learning activities? Third, what specific factors in active learning classrooms influence students' learning activities? Fourth, what is the students' satisfaction with the active learning classroom environments? Fifth and finally,

what needs and preferences do students have for informal learning spaces? Before the interviews began, the students interested in participating were recorded and screened, aiming for an even distribution of students interviewed in terms of major, grade level, and the type of classroom they were in. Fifteen participants were finally identified for the interview in this study. Table 4 presents the main background characteristics of the students interviewed, and these interviewees were able to cover as many grades, majors, and disciplines as possible at this university with a high degree of statistical significance. The interview time for each participant was controlled to be between 10 and 20 min. After the full transcript of the interview, the texts were categorized and analyzed in depth using the programmed grounded theory proposed by Strauss and Corbin [58], and the interview materials were computer coded using Nvivo qualitative analysis software. The interview content was processed through three levels of coding. Open coding (level 1 coding) focused on the material itself, assigning various themes to the meaning expressed in the interview content and categorizing it according to its attributes [57]. According to the relationship between each code (causal, time, semantic, situational, etc.) to connect and cluster, the axial coding (level 2 coding) was formed. The axial coding could be summarized and integrated using the core theme and finally form the selective coding (level 3 coding). The coding process was conducted separately by the corresponding author of this paper and another independent coder and then corrected by multiple rounds of discussion to improve the internal confidence.

**Table 4.** Background characteristics of the students who participated in the interviews.

| Classroom Type | Numbers | Academic Stage | | Gender | | Professional Discipline | | |
| --- | --- | --- | --- | --- | --- | --- | --- | --- |
| | | Undergraduate | Graduate | Male | Female | Social Science | Engineering | Science |
| Table and chair integrated | 2 | 1 | 1 | 0 | 2 | 1 | 1 | 0 |
| Individual splicing combination | 5 | 2 | 3 | 3 | 2 | 1 | 2 | 2 |
| Multi-person splicing combination | 4 | 3 | 1 | 2 | 2 | 1 | 2 | 1 |
| Multi-person fixed combination | 4 | 4 | 0 | 1 | 3 | 3 | 1 | 0 |

Note: Due to the limitation of the interview sample size, the participants' educational levels in this study were only distinguished between undergraduate and graduate students, and they were not subdivided in terms of grade level.

## 4. Results

### 4.1. The Current State of Student Perceptions of Active Learning Classrooms Based on Informal Learning Experiences

Based on the informal learning experience, this study conducted a descriptive statistical analysis of the questionnaire scale of students' perceptions of active learning classrooms, and the results are shown in Table 5. In active learning classrooms, the overall satisfaction of students' informal learning experiences was 4.15, which was between "relatively satisfied" and "very satisfied", indicating that the overall satisfaction level of the students' informal learning activities in active learning classrooms was good. Among the environmental dimensions of active learning classrooms, the students were most satisfied with the informal learning experience of "furniture perception", followed by the dimensions of "instructional interaction" and "physical environment", and were least satisfied with the informal learning experience of "learning support". Regarding the standard deviation, the "physical environment" dimension of active learning classrooms had the smallest value, indicating less dispersion among individuals in the group and that students' perceptions of it were relatively consistent. The "learning support" dimension of active learning classrooms had the largest value, indicating that the degree of dispersion between individuals in the group was large and the students' perceptions of it were relatively different.

**Table 5.** Student satisfaction statistics in active learning classrooms from the informal learning perspective.

| Dimensions | Mean | Standard Deviation |
|---|---|---|
| Instructional Interaction | 4.27 | 0.582 |
| Furniture Perception | 4.29 | 0.651 |
| Learning Support | 3.81 | 0.796 |
| Physical Environment | 4.24 | 0.579 |
| Overall Satisfaction | 4.15 | 0.645 |

Note: The larger the value of the mean, the higher the student satisfaction is, and the smaller the value of the mean, the lower the student satisfaction is.

### 4.2. The Critical Environmental Elements Affecting Students' Perceptions of Informal Learning in Active Learning Classrooms

In this study, exploratory factor analysis was conducted on the questionnaire of students' perceptions in active learning classrooms based on the informal learning experience to reduce the scale data, simplify the data analysis, and verify the validity of the scale structure. The KMO test and Bartlett's spherical test analyzed the scale items, and the KMO test result was 0.91, which was bigger than 0.9, and the Sig of Bartlett's spherical test took a value of 0.000, both indicating that the scale data were suitable for factor analysis. After reclustering, Table 6 shows the spatially influential factor division of the active learning classrooms after reclustering. After several explorations, one question item with a factor component coefficient less than 0.5 was finally excluded, and four factor components with eigenvalues greater than 1 were obtained, while the cumulative contribution of the reclustered factor principal components was 70.837%. According to the initial dimension design of the scale, Factor 1 is related to the attributes of tables and chairs and spatial attributes of the active learning classroom, so factor 1 can be named the "spatial perception" dimension. Factor 2 is related to the active learning classroom's sound, light, thermal, and decorative environments, so factor 2 can be named the "physical environment" dimension. Factor 3 is related to the display and interactive devices in the active learning classroom, so factor 3 can be named the "interactive learning" dimension. Factor 4 is related to the user-friendly facilities and storage space in the active learning classroom, so factor 4 can be named the "learning support" dimension. The name of each factor was replaced by "ALC_F1", "ALC_F2", "ALC_F3," and "ALC_F4," respectively.

In order to control the effects of demographic variables such as student gender, academic stage, major discipline, and active learning classroom type on the regression analysis, independent sample *t*-tests were conducted for student gender, and one-way ANOVAs was conducted for the student academic stage, major discipline, and active learning classroom type, respectively, and it was found that none of these demographic data had a significant effect on this linear regression model.

In order to explore the critical spatial elements that influence students' perceptions of informal learning in active learning classrooms and establish regression equations, this study conducted a linear regression analysis between the reclustered spatial perception factors of active learning classrooms and students' overall satisfaction with informal learning. This linear regression yielded a Durbin–Watson value of 1.497, indicating the good independence of the students who participated in the study. Second, the VIFs between the independent variables involved in this linear regression model were all less than three, indicating that there was no multicollinearity in the data. As shown in Table 7, the linear regression model fit well, with $R^2 = 0.658$, indicating that the four active learning classroom spatial perception factors collectively explained 65.8% of the variance in their overall satisfaction with informal learning. Among them, the "spatial perception" and "learning support" factors could significantly affect the students' overall satisfaction with informal learning. (All *p* values were less than 0.001, and the Beta values were 0.324 and 0.286, respectively.) The regression equation is as follows: The overall informal learning satisfaction = 0.253 + 0.342 × ALC_F1 + 0.231 × ALC_F4.

**Table 6.** Spatial factorization of active learning classrooms after reclustering.

| Factors | Question Items | 1 | 2 | 3 | 4 |
|---|---|---|---|---|---|
| Spatial perception ALC_F1 | Space comfort in the classroom | 0.771 | | | |
| | Comfortable use of tables and chairs | 0.764 | | | |
| | Spatial flexibility in the classroom | 0.759 | | | |
| | Area per person in the classroom | 0.723 | | | |
| | Spatial diversity in the classroom | 0.710 | | | |
| | Flexibility of use of tables and chairs | 0.706 | | | |
| | Usable area of tables and chairs | 0.699 | | | |
| | Equality of space layout in the classroom | 0.523 | | | |
| Physical environment ALC_F2 | Ventilation in the classroom | | 0.829 | | |
| | Artificial lighting in the classroom | | 0.796 | | |
| | Natural lighting in the classroom | | 0.793 | | |
| | Temperature and humidity in the classroom | | 0.744 | | |
| | Color scheme in the classroom | | 0.724 | | |
| | Classroom decoration style | | 0.682 | | |
| Interactive learning ALC_F3 | Movable writing whiteboard in the classroom | | | 0.777 | |
| | The use of multi-screen monitors in classroom | | | 0.699 | |
| | Interactive software experience in the classroom | | | 0.682 | |
| | Clarity of electronic displays in the classroom | | | 0.576 | |
| Learning support ALC_F4 | Storage space in the classroom | | | | 0.754 |
| | Power outlets in the classroom | | | | 0.749 |
| | WiFi signal in the classroom | | | | 0.564 |

Extraction method: principal component analysis; rotation method: Kaiser normalized maximum variance method. The rotation converged after 7 iterations.

**Table 7.** Linear regression between active learning classroom spatial factors and overall satisfaction.

| Variables | B | SE | Beta | T | Sig |
|---|---|---|---|---|---|
| (Constant) | 0.253 | 0.248 | | 1.017 | 0.311 |
| ALC_F1 | 0.342 | 0.085 | 0.324 | 4.008 | 0.000 *** |
| ALC_F2 | 0.144 | 0.079 | 0.130 | 1.825 | 0.070 |
| ALC_F3 | 0.219 | 0.073 | 0.216 | 2.992 | 0.003 |
| ALC_F4 | 0.231 | 0.049 | 0.286 | 4.676 | 0.000 *** |
| R = 0.811 | $R^2$ = 0.658 | Adjusted $R^2$ = 0.649, F = 71.749 *** | | | |

*** $p < 0.001$. Variables: (constant), ALC_F1 = "spatial perception", ALC_F2 = "physical environment", ALC_F3 = "interactive learning", and ALC_F4 = "learning support". The dependent variable is the overall informal learning satisfaction with active learning classrooms.

*4.3. Strengths and Weaknesses of Active Learning Classroom Environments from the Perspective of Students' Informal Learning*

This study used the three-level coding of grounded theory to analyze the interview data, and 146 key points were obtained through the first-level coding. The second-level coding was conducted according to the relationship between the first-level codes, and 10 second-level codes were summarized, including technical support, supporting facilities, physical environment, space perception, furniture design, learning purpose, learning atmosphere, learning activities, environmental status, and improvement measures. Then, the third-level coding was conducted according to the internal relationship of the 10 second-level codes, and 3 core influencing factors were obtained—the space dimension, student dimension, and management dimension—as shown in Table 8.

The frequencies and percentages of the 10 second-level coding categories and 3 third-level coding core factors were obtained in the coding process, as shown in Table 9. Among them, "spatial perception" (19.80%), "furniture design" (12.94%), "improvement measures" (12.43%), and "learning atmosphere" (11.85%) were important secondary coding categories affecting the informal learning activities in active learning classrooms, while the "spatial

dimension" was the important core element in the evaluation process of active learning classrooms from the perspective of students' informal learning.

**Table 8.** Three-level coding of students' perceived evaluation of informal learning in active learning classrooms.

| Level 1 Code | Level 2 Code | Level 3 Code |
|---|---|---|
| "The electronic monitors in the classroom work well". | Technical support | Space dimension |
| "Active learning classrooms have more power outlets". "Active learning classrooms have more storage space". "Active learning classrooms are convenient for e-learning". | Supporting facilities | |
| "The lighting in the active learning classroom is very bright". "The air conditioning in the active learning classroom is very good". "The interior decoration of the active learning classroom is very good". | Physical environment | |
| "Active learning classroom space is more private". "Active learning classroom area is small". "Active learning classroom space is very self-controlled". "Active learning classroom can do flexible space separation". "Active learning classroom can place a lot of things". "Active learning classroom can check the classroom self-study status at any time". "Active learning classroom can check the course class status at any time". "Active learning classroom space has a sense of security". "Active learning classroom space is less dense". | Space perception | |
| "The seats and furniture in the active learning classroom are movable". "The desks shake when used". "The design of the seats is very user-friendly". "The materials of the seats are comfortable". "The desks have a large usable area". "The seats have a certain chance of being damaged". | Furniture design | |
| "The need to finish writing papers". "The need to review for final exams". "The need to wait for upcoming classes". | Learning purpose | Student dimension |
| "Collaborative group learning environment". "Individual independent learning environment". | Learning atmosphere | |
| "Studying for online courses". "Reviewing exam content". "Pre-learning what I will study". "Completing after-class assignments". "Completing essay writing". "Studying my expertise on my own". "Taking breaks between studies". | Learning activities | |
| "Classroom space availability time in active learning classrooms". "Proximity and convenience of active learning classrooms". | Environmental status | Management dimension |
| "Increase the number of active learning classrooms built". "More timely management". "Increase the opening hours of active learning classrooms". "Improve the comfort of furniture". "Upgrade electronic display equipment". "Improve air conditioning systems". "Increase storage space". "Increase the number of outlets". | Improvement measures | |

By analyzing the categories and core factors of the coding, as well as the frequency and percentage of second-level and third-level coding, this study could yield the following three findings from the perspective of the students' experiences with informal learning use:

(1) Suitable spatial perception

The spatial dimension (43.26%) was the largest core factor in the interviews, and it reflected the most intuitive perception of students' experiences in active learning classrooms. First, the spatial perception of the classroom was an important influencing factor (19.80%) in students' choice of active learning classrooms as a place for informal learning. For example, most students believed that active learning classrooms were more spatially private or that active learning classrooms were more spatially self-controlled, and so on. Second, many students were satisfied with the furniture design of the active learning classrooms (12.94%),

namely with larger desks, comfortable seating, and movable furniture attributes that are more adapted to informal learning activities. Third, the supporting facilities (5.33%) and physical environment (4.98%) of the active learning classroom also impacted students' informal learning experiences, and the students expressed relatively positive perceptual feedback concerning them.

(2)     Positive learning atmosphere

The student dimension (24.87%) was the most relevant result for exploring students' perceptions of informal learning. First, the students considered the learning atmosphere to be the main factor influencing informal learning activities in active learning classrooms (11.85%), and many considered active learning classrooms suitable for independent informal learning activities. Second, the students' learning activities in active learning classrooms (7.37%) were more diversified and abundant than in traditional classrooms, such as online course learning, essay writing, or course assignments. Third, the students' learning purpose (5.65%) also affected informal learning activities, such as reviewing exams, waiting for the curriculum, and finishing papers.

(3)     Relative lack of resource management

The management dimension of active learning classrooms (18.98%) is also a critical factor influencing students' informal learning. First, the improvements proposed for active learning classrooms (12.43%) could indicate poor spatial perception in students' informal learning experiences, such as timelier logistical management, increasing the number of active learning classrooms built, and increasing the number of electrical outlets in classrooms. In addition, the current state of the active learning classroom environment (6.55%) is also a factor that influences whether students choose active learning classrooms for informal learning activities, such as the availability of space in active learning classrooms and the proximity of active learning classrooms to student dormitories.

**Table 9.** Frequency and percentage of perceived evaluations in active learning classrooms from the informal learning perspective.

| Core Factors | Category | Second-Level Frequency | Second-Level Percentage | Third-Level Frequency | Third-Level Percentage |
|---|---|---|---|---|---|
| Space dimension | Technical support | 1 | 0.21% | 69 | 43.26% |
| | Supporting facilities | 7 | 5.33% | | |
| | Physical environment | 15 | 4.98% | | |
| | Space perception | 21 | 19.80% | | |
| | Furniture design | 25 | 12.94% | | |
| Student dimension | Learning purpose | 9 | 5.65% | 49 | 24.87% |
| | Learning atmosphere | 16 | 11.85% | | |
| | Learning activities | 24 | 7.37% | | |
| Management dimension | Environmental status | 8 | 6.55% | 28 | 18.98% |
| | Improvement measures | 20 | 12.43% | | |

## 5. Discussion

As a new learning environment in the higher education learning space continuum, active learning classrooms increasingly influence students' learning experiences. Improving students' informal learning experiences and perceptions in active learning classrooms is critical to building a full-time sustainable education environment. By distributing questionnaires to students and conducting semi-structured interviews, as well as using descriptive statistics, linear regression, and grounded theory methods to process the obtained scale data and interview transcripts, this study concludes the status of students' satisfaction with the current active learning classroom environment as a place for informal learning, as well as the critical active learning classroom environment factors that can influence students'

informal learning experiences and perceptions, with the aim of suggesting certain design strategies for subsequent active learning classroom optimization and enhancement.

### 5.1. Students Were Most Satisfied with the Furniture Design Dimension of the Active Learning Classroom and Least Satisfied with the Learning Support Dimension

As a result of sustainable development education [59], next-generation learning spaces [30], intelligent teaching environments [60], and other educational concepts, active learning classrooms have changed and innovated to a certain extent compared with traditional lecture-based classrooms in terms of learning concepts, design principles, form features, and usage methods [61]. In terms of furniture design, the active learning classrooms have completely changed from the regular and fixed furniture form of the previous classrooms to a more autonomous, inclusive, cooperative, and flexible principle, giving students more diverse and creative furniture use scenarios. This abundant and diverse furniture design fits well with students' self-initiated, self-regulated, and self-responsible informal learning activities. Various types of furniture designs, such as the integrated type of tables and chairs, the single-person spliced combination type, the multi-person spliced combination type, and the multi-person fixed combination type, can meet the diverse learning forms and learning purposes of informal learning, such as electronic collaborative communication, interactive sharing of ideas, individual independent reading and learning, and the interaction of teamwork. In this study, the students' satisfaction with the movable, diverse, and comfortable furniture design of active learning classrooms reached a value of 4.29, which was the most satisfying spatial element of the students' informal learning process. Parsons (2016) found in a study that spliceable combinations of semicircular desks in active learning classrooms had a more positive impact on students' usual learning communication and interaction and was able to form different patterns of furniture combinations, depending on the students' activities [62]. Yeoman et al. (2019) argued that by changing the furniture design in the room, different categories of informal learning activities could be supported, including student-centered learning, individual learning, and collaborative learning, so more flexible furniture could build a more active informal learning environment [49].

The design principles of active learning classrooms are based on new learning theories such as active learning, collaborative learning, and deep learning, and their main emphasis is on the diversification of learning spaces and the enrichment of technological equipment. Although the more flexible and diverse environment of active learning classrooms has already had a more positive impact on students' learning experiences, the lack of humane facilities in classroom spaces still exists, such as insufficient storage space, uneven distribution of power outlets, and unstable network signals. However, this overemphasis on learning styles and activities often neglects students' most basic personal use needs. In this study, the students' satisfaction with the learning support facilities in active learning classrooms was only 3.81, which was the least satisfactory element of the space for the students' informal learning process. Porterfield et al. (2020) found that adding personal storage space to active learning classrooms or including an adequate number of electrical outlets could better facilitate student learning activities [63], and Robert et al. (2015) found that although students rated active learning classrooms positively, the lack of user-friendly amenities in classrooms could also lead to negative student learning experiences [64]. This finding was also reflected in the student interviews, where 8 of 15 respondents (53.3%) felt that the storage space in the active learning classroom needed to be increased. Twelve students (80%) felt that the number of electrical outlets in the active learning classroom was low and wanted more to be added. It is worth noting that the need for active learning classroom learning support elements is generally higher in the upper grades (students in their senior year and above) than in the lower grades (students in their junior year and below) due to more e-learning or team-based learning in the upper grades, such as online classes, writing papers, and organizing the contents of electronic reports, as well as a more homogeneous learning style in the lower grades. However, according to the interviews,

the students in the early grades also preferred the active learning classroom as the place for their extracurricular learning due to its more comfortable indoor environment, more convenient learning support facilities, and the ability to work with multiple students to meet their personalized learning needs.

*5.2. The Spatial Perception and Learning Support Dimensions of Active Learning Classrooms Are the Critical Factors Influencing Students' Informal Learning Experiences*

Informal learning, as an unstructured act of active learning by students [6], not only enriches the way students learn but also contributes, to a great extent, to their social and cultural engagement [16]. With the development of sustainable education, more informal learning has been generated on campus, and students have gradually become the dominant learners [8]. A large amount of learning outside of class time has become a new way for students to acquire knowledge. In addition to serving as a place where formal learning occurs for students, active learning classrooms can also meet the contextual, collaborative, self-directed, and flexible characteristics needed for students' informal learning. In this study, the linear regression between the environmental dimensions of the active learning classroom and students' overall satisfaction with the informal learning experience showed that flexible spatial perception is the critical factor affecting the overall satisfaction with the informal learning experience in the active learning classroom ($p < 0.001$, Beta = 0.324). In addition, compared with other learning environments, the comfortable and flexible spatial perception of active learning classrooms can also influence students' informal learning more effectively for the following reasons. First, there is the more comfortable spatial environment. For example, one respondent said, "It is comfortable to study in an active learning classroom, the environment is great, and a good spatial environment can motivate greater motivation to learn". Second, there is the flexible and variable layout of the individual study space. One student explained this by saying, "The classroom space can be changed at the will, enabling the formation of multiple independent individual study spaces". Third, the layout of the space supports multi-person interaction and group practice. Some respondents said, "The classroom space is particularly suitable for group collaboration, and the form of space that can be put together and combined helps us broaden our ideas and share everyone's knowledge and ideas". The findings of some researchers can also confirm this point. Oliveira et al. (2016) found that when the overall spatial environment of the learning space was not comfortable or attractive, it would reduce the time for students to study there [65]. Granito et al. (2016) showed that students prefer flexible classroom space layouts because they can flexibly switch between different learning styles [66].

Education in the 21st century is inseparable from the support of information technology. Convenient and portable computer equipment and electronic learning spaces covered by a wireless network are important factors in improving students' learning satisfaction. The learning styles of contemporary college students are more diversified and diverse, such as learning through the interactive use of multiple electronic devices, interactive team learning with network connectivity, and visualized experiential learning, all of which require rapid development and responsiveness from the learning environment. As a space for formal and informal learning, the active learning classroom should pay more attention to the combination of physical and virtual as well as humanized and informational to give students a more comprehensive and holistic environment to support their learning. Although student satisfaction with the learning support elements of current active learning classrooms is low, a linear regression based on the relationship between environmental factors in active learning classrooms and the overall satisfaction with students' informal learning experiences can reveal that the learning support dimension of active learning classrooms remains a critical spatial element influencing students' informal learning experiences ($p < 0.001$, Beta = 0.286). Its main performance is: First, the available storage space. Active learning classrooms have increased the amount of usable space per person, and there are more areas in the classroom where personal belongings can be placed, with some

respondents stating, "There is more space in the classroom to place personal belongings properly". Second, there is the wireless coverage of WiFi devices. Active learning classrooms are designed with the addition of ICT in mind, and their network environment is relatively more complete. Some interviewees mentioned, "The network signal of the active learning classroom is stable, and they usually choose this place to complete their online courses". Third, there are evenly distributed power outlets. Active learning classrooms are designed with evenly distributed power outlets on all four walls of the classroom to facilitate the power needs of students' information-based learning. Some interviewees said, "More adequate power outlets make it easier to use computers and other electronic-based devices, and enable longer learning time".

*5.3. The Spatial Privacy and Learning Atmosphere of Active Learning Classrooms Can Promote Informal Learning for Students*

Informal learning emphasizes the social, diverse, collaborative, and purposeful nature of learning compared with formal learning [36]. Therefore, a learning environment that focuses more on spatial privacy is more effective for students' informal learning [44]. Students welcome the relatively more private learning spaces of active learning classrooms for informal learning, and their quiet and comfortable spatial environment and free and abundant spatial layout can facilitate informal learning styles such as individual study, collaborative discussions, gatherings and meetings, retreats, and readings [40] while also supporting students' choice of learning activities according to their needs and preferences [67] and increasing the frequency of use [41]. Of the 15 respondents who participated in the one-on-one interviews, 11 students (73.3%) believed that the spatial privacy of the active learning classroom had a more positive impact on informal learning activities than other informal learning spaces, as evidenced by the following. First, a high-quality physical environment creates better privacy. For example, some respondents said, "The classroom is a quiet and private learning environment with good sound insulation in the indoor environment as well as walls for shielding". Second, the flexible space layout allows for a separate learning environment. One student stated, "When other people are studying in the classroom, I can move the furniture to an unoccupied corner to study and avoid the interference of others". A study by Deng et al. (2017) showed that most students prefer to study in a quiet and solitary environment so that the quiet learning environment allows them to concentrate on their reading and study fully [68]. Beckers et al. (2016) found that students prefer to choose seats in remote corners for informal learning to avoid the distractions brought by other people [50].

Active learning classrooms have a stronger learning atmosphere than other learning spaces. In formal learning spaces, students experience them through their senses and give them meaning for learning, and the students become more familiar with the classroom space. In addition, active learning classrooms often have adequate learning furniture and equipment, and they also give users a certain degree of self-control while allowing students to develop a positive sense of spatial belonging, which are the spatial elements that bind students together and build a stronger learning atmosphere. In this study's interviews, seven students (46.6%) mentioned the learning atmosphere of the active learning classroom and believed that its spatial atmosphere was closely related to learning activities. For example, some students mentioned that they "prefer to come to the active learning classroom with their classmates for extracurricular learning to monitor and promote each other". "The active learning classroom has a high learning atmosphere, and seeing everyone's learning behavior can improve one's motivation". In studies such as those by Harrop et al. (2013) and Waldock et al. (2016), it was found that students, both academically and socially, learn near friends and peers, creating an atmosphere of a learning community where everyone can work together, and students can develop a sense of enrichment that "I came here to learn and my friends are already here, so I joined them", meaning that students are more motivated to work in a shared learning environment [67,69].

*5.4. Better Resource Management Helps Students Have a Higher-Quality Informal Learning Environment in Active Learning Classrooms*

The active learning classroom uses many new technology devices, and the layout of the space is also more abundant and flexible to meet the diverse learning needs of students. More accessible resources and available space also require more time management and support, including the proximity of classroom locations, the length of open hours, the number of locations, and the management and maintenance of learning resources. These dimensions of active learning classroom management have a substantial role in enhancing the effectiveness of learning environments, but the current theoretical and practical research on active learning classrooms and informal learning has not received the corresponding attention. Among the 15 interviewees, 13 students (86.6%) mentioned the management status and subsequent improvement of active learning classrooms, which are mainly reflected in three aspects. First, there is the more timely classroom management. Some respondents mentioned, "The classroom has some leftover items and trash after class, which can disturb other students studying". Other respondents also mentioned, "The furniture and technology in active learning classrooms should be managed more frequently; some furniture storage spaces are dirty, and some equipment is damaged and needs to be disposed of on time to ensure an efficient learning environment". Second, there are the longer open hours. Ten of the students who participated in the interviews (66.7%) mentioned their willingness to engage in longer learning activities in the active learning classroom. Some students responded, "While active learning classrooms are preferred for extracurricular learning, their opening hours are somewhat limited and hinder extracurricular learning". Another student mentioned, "Active learning classrooms are comfortable and the equipment is smarter, but they are also more strictly managed and open for shorter periods of time than traditional classrooms, while I would like to study in the active learning classroom for longer periods of time, the reality is often the opposite". Third, there is the more accessible classroom space. One student explained this by saying, "Active learning classrooms are great for studying, but I would also choose other traditional classrooms closer to the dorms". "We would choose a study space closer to us, even if its space use is not as good as an active learning classroom". Overall, the interviews with students revealed that students are willing to spend more time in the active learning classroom for informal learning activities due to its better environment, comfortable layout, variety of spatial features, and abundant configurations. Therefore, the logistic management, equipment maintenance, classroom open hours, and classroom accessibility should be reorganized accordingly.

## 6. Conclusions and Limitations

With the flourishing research and practice of active learning classrooms in recent years, its changes and innovations in learning styles, teaching models, spatial environments, and educational technologies have profoundly impacted sustainable education development. The Education 2030 Framework for Action [15], released by OECD in 2015, defines the importance of lifelong education and full-time education and emphasizes that students' informal learning is "valuable learning", being as valuable as formal learning. As a new learning environment for sustainable education, active learning classrooms have emerged in relevant theoretical and practical studies [54,59]. However, there is no academic consensus on the use and theory building of informal learning in active learning classrooms. Based on students' perspectives, this study explores the impact of the sustainable educational environment in the active learning classroom on students' informal learning perception. Using a combination of quantitative and qualitative empirical analyses, this study found that first, students were more satisfied with their informal learning experiences in active learning classrooms, and most students preferred active learning classrooms as their informal learning environment. Second, flexible and comfortable space perception and humanized learning support facilities in active learning classrooms are critical spatial factors affecting students' informal learning. Third, a private environment and sufficient

learning atmosphere in active learning classrooms can promote students' informal learning ability. Fourth, for active learning classrooms, better resource management helps them form a better-quality and full-time learning environment.

The findings mentioned above have the following implications for the optimal design and sustainable development of active learning classrooms in the future.

First, active learning classrooms should actively experiment with more diverse and flexible space layouts as well as emphasize more comfortable and student-centered environmental perception designs, such as by using more acoustic and soft-colored interior materials to improve students' spatial comfort, increasing the usable area of classroom chairs and tables to improve students' usage comfort, designing a variety of flexible tables and chairs to meet different learning activities such as individual and multi-person learning, expanding more areas for leisure and learning in classrooms, and designing electronic interactive devices that can meet both the needs of the curriculum and the needs of students' extracurricular learning activities.

Second, the active learning classroom should improve and increase more diverse and convenient humanized facilities, such as through multiple lockers for personal use in active learning classrooms, floor outlets evenly spaced on the floor that can be hidden for storage, WiFi signal boosters that can be used by dense numbers of people, retail vending machines for students who use the classroom for long periods of time, and information interaction software that students can use independently.

Third, the environmental design of active learning classrooms should focus on the privacy of the space and the establishment of a learning atmosphere, such as by designing some mobile partitions that can enrich the space level so as to form a diversified personal learning space from a private one semi-open to the informal learning use of the classroom, designing a more integrated and wrapped furniture design to meet the personal privacy needs of students when they study independently after class, and increasing learning service facilities such as mobile whiteboards or fixed interactive whiteboards to encourage students' extracurricular team communication and cooperation and improve students' control over the sound, light, and thermal environment of the classroom to meet students' needs for a learning space atmosphere.

Fourth, the timeliness and accessibility of logistical and resource management in active learning classrooms can be improved by, for example, extending the available time of the classroom, increasing the frequency of equipment inspections and indoor cleaning of the classroom, increasing the space supply of the active learning classroom in extracurricular time, repairing and maintaining various technical equipment and furniture facilities in the classroom in a more timely manner, and making the site of the active learning classroom closer to the dormitory and canteen for students.

Student-centered active learning classrooms are designed primarily to promote active and deep learning, which requires a higher level of student engagement. By improving the environmental design and experiential perception of the learning space, it can effectively promote students' learning involvement [70]. In addition, similar to formal learning, learning engagement in students' informal learning behavior is also an intermediary variable to improve students' abilities of active learning and deep learning. The more frequent students' informal learning behavior is, the more they will participate in learning [71], and this kind of participation behavior will also increase students' active learning ability. In summary, the construction and development of learning environments are closely related to students' learning perceptions. By rethinking students' informal learning experiences and perceptions, the high-quality development of active learning classrooms can be more comprehensively enhanced, thus enhancing students' full-time learning experiences, which in turn promotes student learning engagement and active learning in active learning classrooms. In addition, as a new type of learning environment for sustainable education development, active learning classrooms should not only meet the learning theories and educational strategies for formal learning but should also focus on and enhance the scenarios used by students for informal learning. This holistic classroom environment

that combines formal and informal learning creates a full-time learning environment that promotes student active learning to achieve the goal of quality and sustainable higher education development.

This study also has certain limitations due to research capacity and time constraints. First, the experimental data come from students' self-reporting, which may affect the authenticity of the data due to discrepancies in students' self-perceptions, and research methods such as objective measurement and performance evaluation can be integrated in the future. Second, the time of the study was relatively limited, and this study chose data from one semester. In the future, a more in-depth and reasonable exploration can be conducted by tracking research on students' long-term learning space use processes. Third, the object of the study was relatively singular, and only one university was selected for the collection of student data and classroom models in this study. In the future, more diverse studies can be conducted by investigating several universities with different disciplinary and professional characteristics at the same time, which makes the research findings more convincing and generalizable.

**Supplementary Materials:** The following supporting information can be downloaded at: https://www.mdpi.com/article/10.3390/su14148578/s1, Spatial satisfaction in two types of classrooms_Raw Data.

**Author Contributions:** Conceptualization, S.J. and Y.G.; methodology, L.P., Y.D. and S.J.; software, S.J.; validation, L.P., Y.D. and S.J.; formal analysis, S.J.; investigation, Y.D. and S.J.; resources, L.P., Y.D. and S.J.; data curation, S.J.; writing—original draft preparation, S.J.; writing—review and editing, L.P. and S.J.; visualization, Y.D. and S.J.; supervision, L.P. and S.J.; project administration, L.P.; funding acquisition, L.P. All authors have read and agreed to the published version of the manuscript.

**Funding:** This research was funded by the National Natural Science Foundation of China, grant number 51978294.

**Institutional Review Board Statement:** This study was conducted according to the guidelines of the Declaration of Helsinki and approved by the Medical Ethics Committee, Tongji Medical College, Huazhong University of Science and Technology, authorization number: 2022-S046.

**Informed Consent Statement:** Informed consent was obtained from all subjects involved in the study.

**Data Availability Statement:** The data used to support the findings of this study are included within the article.

**Acknowledgments:** The authors would like to thank the teachers and students at Huazhong University of Science and Technology for their support and encouragement.

**Conflicts of Interest:** The authors declare no conflict of interest.

## Appendix A. Questionnaire Template

*Questionnaire Survey on Satisfaction with the Informal Learning Experience in Active Learning Classrooms at Huazhong University of Science and Technology*

Hello! Thank you very much for taking the time to fill out this questionnaire. This questionnaire aims to ascertain students' informal learning satisfaction with the classroom space and serve as a resource for optimizing classroom spaces in universities.

1. The classroom number you are currently in is:
2. Your gender is:

   A. Male                              B. Female

3. Your academic stage is:

   A. Undergraduate
   B. Master's degree
   C. PhD
   D. Other

4. Your professional discipline is:

    A. Philosophy, economics, and law
    B. Education, literature, and history
    C. Science, engineering, agriculture, and medicine
    D. Military science, management, and art

5. How satisfied are you with the following elements of active learning classrooms at HUST when you study informally?

| (1 = Very dissatisfied; 2 = Dissatisfied; 3 = Neutral; 4 = Satisfied; 5 = Very satisfied) | | | | | |
|---|---|---|---|---|---|
| **Title** | **1** | **2** | **3** | **4** | **5** |
| Instructional Interaction | | | | | |
| Clarity of electronic displays in the classroom | ☐ | ☐ | ☐ | ☐ | ☐ |
| The use of multi-screen monitors in the classroom | ☐ | ☐ | ☐ | ☐ | ☐ |
| Movable writing whiteboard in the classroom | ☐ | ☐ | ☐ | ☐ | ☐ |
| Interactive software experience in the classroom | ☐ | ☐ | ☐ | ☐ | ☐ |
| Space comfort in the classroom | ☐ | ☐ | ☐ | ☐ | ☐ |
| Spatial flexibility in the classroom | ☐ | ☐ | ☐ | ☐ | ☐ |
| Spatial diversity in the classroomm | ☐ | ☐ | ☐ | ☐ | ☐ |
| Equality of space layout in the classroom | ☐ | ☐ | ☐ | ☐ | ☐ |
| Furniture Perception | | | | | |
| Area per person in the classroom | ☐ | ☐ | ☐ | ☐ | ☐ |
| Usable area of tables and chairs | ☐ | ☐ | ☐ | ☐ | ☐ |
| Comfortable use of tables and chairs | ☐ | ☐ | ☐ | ☐ | ☐ |
| Flexibility of use of tables and chairs | ☐ | ☐ | ☐ | ☐ | ☐ |
| Learning Support | | | | | |
| Storage space in the classroom | ☐ | ☐ | ☐ | ☐ | ☐ |
| Power outlets in the classroom | ☐ | ☐ | ☐ | ☐ | ☐ |
| WiFi signal in the classroom | ☐ | ☐ | ☐ | ☐ | ☐ |
| Physical Environment | | | | | |
| Sound insulation in the classroom | ☐ | ☐ | ☐ | ☐ | ☐ |
| Natural lighting in the classroom | ☐ | ☐ | ☐ | ☐ | ☐ |
| Artificial lighting in the classroom | ☐ | ☐ | ☐ | ☐ | ☐ |
| Temperature and humidity in the classroom | ☐ | ☐ | ☐ | ☐ | ☐ |
| Ventilation in the classroom | ☐ | ☐ | ☐ | ☐ | ☐ |
| Classroom decoration style | ☐ | ☐ | ☐ | ☐ | ☐ |
| Color scheme in the classroom | ☐ | ☐ | ☐ | ☐ | ☐ |

6. What is your overall satisfaction with the space in active learning classrooms at HUST when you study informally?

    A. Very dissatisfied
    B. Dissatisfied
    C. Neutral
    D. Satisfied
    E. Very satisfied

7. Do you have any other suggestions for the active learning classrooms at HUST? Do you have any other ideas or feedback?

**Appendix B. Interview Outline**

*Semi-Structured Interview Outline of the Informal Learning of Students in Active Learning Classrooms at Huazhong University of Science and Technology*

Semi-structured interviews will explore students' experiences and perceptions of using active learning classrooms for informal learning outside of class time and attempt to draw out the impact of the learning theories in active learning classrooms on students' learning activities and technology use.

The main questions and possible follow-up questions are as follows:

1. Your personal information (major and year) and the number of the active learning classroom where you are located.
2. How often do you come to the active learning classroom to study outside of class time?
   - Your frequency of extracurricular time in the active learning classroom is?
   - When do you usually come to the active learning classroom?
3. What learning activities do you typically perform outside of class in the active learning classroom?
   - Do you use the active learning classroom with your classmates for informal collaborative learning outside of class time? If so, how satisfied are you with collaborative learning in the active learning classroom? What design and environmental factors affect teamwork?
   - Do you choose active learning classrooms for breaks or chats outside of class time?
4. How do you like the learning environment in the active learning classroom?
   - How did you feel the first time you came to an active learning classroom outside of class time?
   - What do you think about the physical environment, furniture design, and technical equipment of the active learning classroom that attracts you to informal learning in the active learning classroom?
   - What do you think is particularly attractive about the active learning classroom? How does it help you during your informal learning sessions?
   - Which aspect of the active learning room do you think you are not satisfied with, or what do you think could be improved?
5. Why did you choose an active learning classroom for informal learning rather than a traditional lecture-based classroom for informal learning?
   - What are some differences between active learning classrooms and traditional lecture-based classrooms?
   - How have these differences you noticed impacted your learning?
   - Did the active learning classroom and traditional lecture classroom environments make a difference in your attention or engagement during informal learning? If so, what do you think might cause these differences?
6. Why do you choose active learning classrooms for informal learning rather than public learning spaces or libraries for informal learning?
   - What differences do you notice between active learning classrooms and public spaces and libraries?
   - How have these differences you noticed impacted your learning?
   - Does the active learning classroom and public learning environment make a difference in your attention or engagement during informal learning? If so, what do you think may have contributed to these differences?
7. What are your needs and preferences for learning spaces? What else would you like to see in the design?
8. Finally, do you have any comments, suggestions, or feedback about active learning classrooms?

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
