# Peer review of "Students’ Perceptions of Active Learning Classrooms from an Informal Learning Perspective: Building a Full-Time Sustainable Learning Environment in Higher Education"

_sustainability, doi:10.3390/su14148578_

Round 1
Reviewer 1 Report
This is very unusual research at higher education. This research can be very useful for other scientists who have research at education. Specially who engaged in research of teaching aids and facilities.
Some part at research methodology need to move at discussion. The discussion is too theoretical, it should be supported by the results of the research.
List of references is not good. A lot of units on the list are older then 5 or 10 years. No new researches. Authors must refreshes that.
Author Response
Dear Reviewer, Thank you very much for taking time out of your busy schedule to read my manuscript and for your valuable comments on the manuscript, which are very meaningful to me. I have made some corrections and explanations to your suggestions, which can be found in the attached "Response to Reviewer 1 Comments.docx", and some of the issues in the manuscript have been revised. Thank you again for your teaching and suggestions! Sincerely Shitao Jin

Reviewer 2 Report
Thank you for this interesting work. I think that the paper can be improved by providing clearer example of what your active classroom is like? It remains rather as a concept and should be illustrated how these rooms are used and not only how they are designed. In addition, I think that you should also report Eigenvalues and Communalities within Factor Analysis and also report your linear regression following APA-guidelines how to report such an analysis. It does also make no sense to show variance AND SD. Just report SD.
Author Response
Dear Reviewer, Thank you very much for taking time out of your busy schedule to read my manuscript and for your valuable comments on the manuscript, which are very meaningful to me. I have made some corrections and explanations to your suggestions, which can be found in the attached "Response to Reviewer 2 Comments.docx", and some of the issues in the manuscript have been revised. Thank you again for your teaching and suggestions! Sincerely Shitao Jin

Reviewer 3 Report
Dear authors,
First of all, I appreciate the the opportunity to read your article entitled “Students’ Perceptions of Active Learning Classrooms from an Informal Learning Perspective: Building a Full-time Sustainable Learning Environment in Higher Education”, that I have read with great interest.
I would like to say that the topic of this paper is relevant, however, the paper has some points, which should be improved. In my view, the revised version should undergo a new assessment process.
1- The author employed both qualitative and quantitative methodology, this is a good point, however you need to justify why the quantitative methods comes before the qualitative method as regularly the qualitative comes first.
2- I did not see any framework or hypotheses in the study please justify.
3- The author extracted the scale from previous literature (Lei et al., Choi et al., Yang et al.), if these scales is well established, thus no need to conduct EFA.
4- Please discuss the adequacy of sample size
5- You need to support your argument with confirmatory factory analysis to test the scale convergent and discriminant validity showed the composite reliability and the average variance extracted.
6- The theoretical and practical implications need more elaboration
Best wishes
Author Response
Dear Reviewer, Thank you very much for taking time out of your busy schedule to read my manuscript and for your valuable comments on the manuscript, which are very meaningful to me. I have made some corrections and explanations to your suggestions, which can be found in the attached "Response to Reviewer 3 Comments.docx", and some of the issues in the manuscript have been revised. Thank you again for your teaching and suggestions! Sincerely Shitao Jin

Reviewer 4 Report
Thank you so much for inviting me to review the paper entitled “Students’ Perceptions of Active Learning Classrooms from an Informal Learning Perspective: Building a Full-time Sustainable Learning Environment in Higher Education” – paper is quite interesting
Some recommendations are provided:
Abstract – some note how many participants – since survey and interview were done, is this a mixed method study – should also not how the data from survey and interview intersect
Is active learning similar to the concepts of student engagement – can also include on some of the concepts of Astin’s theory of student involvement and Kuh’s student engagement.
With regards to informal learning – why spatial perceptions? There should be some theoretical framework for this. (although provided in the literature review, should have a clearer connection for this) might also include a few statement in the introduction prior to the research objectives
For the factor analysis, did you compute for the variance extracted? Section 4.2 – lines 308 onwards
Would the findings limited to the science and engineering discipline? – table 2
Would the findings reflect that the students from the said department have spent more time in these spaces?
Same with lesser graduate students – would this type of spatial design more suitable for undergrad?
What do you mean by “studying informally”, activities held in the said spaces? By student clubs? Organized by the departments / colleges or self-studying? - line 220
Self-study? Or during extra-curricular activities?
With the learning support the lowest with 3.81 (table 4) but instructional interaction with 4.27, what would this finding mean. Although interactions with faculty (or with peers) is considered high, but student perceived lesser learning occurrence? Did the author further investigate this during the interview?
Regression – did you consider using the background demographics as control variables?
In sum, the paper is quite interesting, some clarity needed on some sections of the paper.
Author Response
Dear Reviewer, Thank you very much for taking time out of your busy schedule to read my manuscript and for your valuable comments on the manuscript, which are very meaningful to me. I have made some corrections and explanations to your suggestions, which can be found in the attached "Response to Reviewer 4 Comments.docx", and some of the issues in the manuscript have been revised. Thank you again for your teaching and suggestions! Sincerely Shitao Jin

Round 2
Reviewer 1 Report
All corrections are good. I do not any comments.
Reviewer 3 Report
i can accept the manuscript in its current form